# The mRNA and microRNA Landscape of the Blastema Niche in Regenerating Newt Limbs

**DOI:** 10.3390/ijms25179225

**Published:** 2024-08-25

**Authors:** Qi Zhang, Bin Lu

**Affiliations:** 1Chengdu Institute of Biology, Chinese Academy of Sciences, Chengdu 610041, China; zhangqi1@cib.ac.cn; 2University of Chinese Academy of Sciences, Beijing 100049, China

**Keywords:** *Cynops orientalis*, limb regeneration, transcriptome, mRNA–miRNA integration, temporal specific expression

## Abstract

Newts are excellent vertebrate models for investigating tissue regeneration due to their remarkable regenerative capabilities. To investigate the mRNA and microRNAs (miRNAs) profiles within the blastema niche of regenerating newt limbs, we amputated the limbs of Chinese fire belly newts (*Cynops orientalis*) and conducted comprehensive analyses of the transcriptome and microRNA profiles at five distinct time points post-amputation (0 hours, 1 day, 5 days 10 days and 20 days). We identified 24 significantly differentially expressed (DE) genes and 20 significantly DE miRNAs. Utilizing weighted gene co-expression network analysis (WGCNA) and gene ontology (GO) enrichment analysis, we identified four genes likely to playing crucial roles in the early stages of limb regeneration: *Cemip*, *Rhou*, *Gpd2* and *Pcna*. Moreover, mRNA–miRNA integration analysis uncovered seven human miRNAs (miR-19b-1, miR-19b-2, miR-21-5p, miR-127-5p, miR-150-5p, miR-194-5p, and miR-210-5p) may regulate the expression of these four key genes. The temporal expression patterns of these key genes and miRNAs further validated the robustness of the identified mRNA-miRNA landscape. Our study successfully identified candidate key genes and elucidated a portion of the genetic regulatory mechanisms involved in newt limb regeneration. These findings offer valuable insights for further exploration of the intricate processes of tissue regeneration.

## 1. Introduction

Many organisms in nature exhibit varying degrees of regenerative capacity, with newts, a group of vertebrates, standing out for their remarkable proficiency in this regard. Newts possess the ability to regenerate a diverse array of organs and tissues, including the limbs, eyes (comprising the lens and retina), and optic nerves, among others [1]. The process of limb regeneration is a highly intricate biological phenomenon that involves the growth and development of multiple tissues such as the epidermis, dermis, limb bone, and skeletal muscle [2]. Broadly, limb regeneration can be categorized into the following phases: wound healing (0–5 days), blastema formation and pattern differentiation, and the differentiation (6 days–20 days), and growth of new cells to reconstruct the limb (21 days–40 days) [1,3]. The unique nature of external limb injuries introduces additional factors, including the immune system response, inflammatory reactions, and cell migration, all of which play crucial roles in the regeneration process [4,5]. Throughout limb regeneration, dynamic changes occur in tissue repair, growth, and other biological processes, collectively forming a complex regulatory network that ensures the normal progression of regeneration [6]. The temporal and differential expression of genes emerges as a crucial factor in regulating the body’s response to external stimuli and the processes of tissue regeneration and development. Consequently, genetic research and analysis of the intricate phenomenon of newt limb regeneration [7,8] represents a prudent approach and a vital avenue for exploring its underlying molecular mechanisms.

MicroRNAs (miRNAs), a class of short, non-coding RNAs, play a crucial role by binding to the 3’ or 5’ end of target mRNAs, thereby inhibiting normal transcription and leading to the reduction or silencing of gene expression [9]. This distinctive feature positions miRNAs as vital regulators in various biological processes. In recent years, significant progress has been achieved in the exploration of co-expression of miRNA and genetic targets in cancer [10,11]. Tumors, characterized by continuous cell division, share certain features with the regenerative processes observed in salamander, including migration, invasion, and extracellular matrix (ECM) remodeling [11,12,13]. Notably, signaling pathways, such as Wnt, Notch, and TGF-β, in both cancer and regeneration, play pivotal roles [6,14,15,16,17]. However, distinctions exist, with some studies suggesting that while both processes share biological processes, regeneration is more ordered and converged compared to the disordered development of cancer [1,18]. Previous studies have aptly described this phenomenon as “similar pathways, different outputs” [19,20]. Considering the critical role of miRNAs in regulating gene expression and their involvement in molecular mechanisms, a comprehensive investigation of limb regeneration through the perspective of miRNA is essential. Previous study has highlighted a significant upregulation of miR-21 expression during axolotl regeneration, underscoring its indispensable role [2,21]. A thorough analysis of differentially expressed miRNAs and their target genes at various regeneration time points is crucial for gaining a deeper understanding of the genetic interactions and regulatory mechanisms underlying this complex biological process.

In this study, we investigated the mRNA and miRNA profiles within the blastema niche during the early stages of limb regeneration in the Chinese fire belly newt (*Cynops orientalis*). Transcriptome sequencing was conducted on the healing tissues at 0 hours, 1 day, 5 days, 10 days, and 20 days post-amputation of the right front limb (Figure 1). Concurrently, miRNA sequencing was carried out to comprehensively capture the molecular changes and regulatory patterns associated with the healing progress. Leveraging the generated sequencing data, we conducted differential expression analysis of genes across different time points, followed by mRNA-miRNA co-expression analysis. Our primary objective is to identify key genes exhibiting differential expression at distinct time points, along with the miRNAs regulating them. By extrapolating from the genetic analysis results, we aim to speculate on the potential significance of these key genes in orchestrating the intricate biological processes involved in newt limb regeneration.

## 2. Results

### 2.1. Identification of Differentially Expressed Genes and miRNAs

Differential expression analysis was conducted across the time course by comparing each time point to the zero-hour control. The analysis revealed 24 significantly differentially expressed (DE) genes and 20 DE miRNAs. Of these, 21 genes and 14 miRNAs were upregulated, while 3 genes and 6 miRNAs were downregulated, as depicted in Figure 2 and Figure 4. Notably, the genes *Cnp*, *Sycn*, *Rbbp8nl*, *Slc6a3*, *Cemip*, and *Rnf207* were consistently differentially expressed across multiple time points, with both *Rbbp8nl* and *Slc6a3* showing significant differential expression in three comparisons (0 h vs. 5 d, 0 h vs. 10 d and 0 h vs. 20 d). This may suggest their significant roles in the early stages of regeneration due to their persistently elevated expression levels following amputation. The gene *Rbbp8nl* plays an indispensable role in double-strand breaks (DSB) [22], while *Slc6a3* is known to mediates dopamine binding and transport.

### 2.2. Co-Expression Modules and Enrichment Analysis

Weighted Gene Co-expression Network Analysis (WGCNA) was employed to identify clusters of genes exhibiting highly interconnected relationships, referred to as modules. Following the normalization and absolute value transformation of gene expression data, genes displaying similar expression trends were grouped into clusters, each representing a distinct module. In this WGCNA analysis, all DE genes were categorized into six co-expression modules based on their expression pattern similarities, each module containing 2 to 5 gene members, represented by distinct colors (Figure 3a). Notably, the blue and green modules demonstrated the strongest significant associations with the time course. However, due to the limitations in non-reference transcriptome annotation, no enriched terms were identified in the green module. All three functional enrichments corresponding to the blue module showed robust statistical significance. Among these, the myoblast migration process, regulated by *Net1* and *Six4*, plays a crucial role in muscle injury and recovery by mediating muscle cells migration [23]. The hyaluronan catabolic process, mediated by *Cd44* and *Cemip*, is crucial for maintaining tissue lubrication, normal structure, and function, and is indispensable in the early stages of tissue repair [24]. Interesting, *Cemip*, also known as *Hybid* or *Kiaa1199*, was significantly upregulated at both 5 and 20 days post-amputation. *Cemip* has been demonstrated to bind hyaluronic acid [25,26], a major component of the extracellular matrix essential for wound repair, cell migration, and skin healing. The ceramide transport process, mediated by *Abca2* and *Cptp*, primarily functions in intercellular signal transduction and apoptosis regulation, modulating the cellular microenvironment in response to external stimuli [27]. These processes create favorable biochemical conditions for limb regeneration post-injury. The statistical significance of their positive correlation with temporal changes further underscores the unique biological significance of this module. Within the yellow module, “cellular extravasation” emerged as the most significant annotation, involving *Bst1*, *Ccl25* and *Crk*. This enrichment is also noteworthy, as cellular extravasation regulates the migration of white blood cells during the inflammatory response and plays a role in lymphocyte activation, setting the stage for the subsequent migration of healing substances, cell growth, and apoptosis [28,29]. The GO enrichment results are shown in Figure 3b, with key genes involved in each biological process provided in Table 1 for a detailed breakdown of the results.

### 2.3. Target Genes of DE miRNAs

The correlation between expression levels of the aforementioned DE genes and miRNAs at various time points is depicted in Figure 4a. Following bidirectional target prediction between miRNA and mRNA, we performed a Pearson correlation test on the DE miRNAs and DE mRNAs to confirm the statistical significance of their correlation. Pairs that met the criteria under the set threshold were selected, resulting in the prediction of 16 DE miRNAs targeting a total of 13 genes. Since miRNAs typically negatively regulate the expression of their target genes, we further refined the predicted target genes. After filtering, only 8 of the predicted target genes met the conditions of being present in the mRNA expression data and exhibiting a significant negatively correlation with DE miRNAs. A correlation network is shown in Figure 4b,c.

Remarkably, one gene, *Rhou*, was targeted by 7 miRNAs, including hsa-miR-21-5p, hsa-miR-210-5p, hsa-miR-150-5p, hsa-miR-194-5p, hsa-miR-141-5p, hsa-miR-1260b, and hsa-miR-155-5p. Notably, *Rhou*, as a non-canonical Wnt-induced gene, has been shown to reduce cell apoptosis and increase proliferation when knocked down [30]. This is attributed to the molecular function of *Rhou*, which regulates cell adhesion, reduces excessive migration, promotes the proper cellular arrangement during regeneration, and regulates cell differentiation and morphological regeneration. Besides *Rhou*, the miRNAs hsa-miR-194-5p and hsa-miR-21-5p also regulate the *Gpd2* gene. The downregulation of *Gpd2* contributes to the reduction of mitochondrial activity and may mediate a metabolic shift from oxidative phosphorylation to glycolysis, which is critical for regenerative tissues [31]. Figure 5 illustrates the potential effects of limb regeneration following the inhibition of expression of *Rhou* and *Gpd2* regulated by specific miRNAs. Consistent with our findings, upregulation of hsa-miR-21-5p expression has also been reported during tissue regeneration processes in the Mexican axolotl (*Ambystoma Mexicanum*) [2]. Through the detection and analysis of target pairs, we constructed a comprehensive co-expression landscape, providing a more intuitive interpretation of the interaction between genes and miRNAs. This serves as data basis for further exploration of the role of miRNAs in newt limb regeneration.

### 2.4. Temporal Dynamics of Expression

As mentioned in the WGCNA analysis, genes with similar functions can be clustered into the same module, which may influence complex biological phenomena. The temporal expression patterns of these genes are also significant interest. Clusters of genes with different temporal expression patterns suggest their activity trends in biological processes during the early stages of regeneration. To enhance clarity in illustrating the temporal changes of key genes, all transcriptome data were categorized into eight clusters, and miRNA data were classified into four categories. Among the three key genes highlighted earlier, *Cemip* was assigned to cluster 3, while *Rhou* and *Gpd2* were associated with cluster 6. The graphical representation (Figure 6a) depicts the temporal dynamics and correlation of these gene and miRNA clusters, underscoring a significant concordance with the observed expression pattern changes of the specified key genes. We also queried the *Pcna* gene, which plays a significant role in several cellular processes related to DNA repair and cell cycle regulation, and serves as a proliferative marker. *Pcna*, which exhibited a potentially periodic expression pattern (though not statistically significant according transcriptome data), was classified into cluster 4 [32]. In addition, our quantitative real-time PCR (qPCR) validations for *Pcna* demonstrated a periodic expression pattern that aligned with transcriptome data, but with significantly reduced expression at both 1 and 10 days post-amputation (Figure 6b).

After analyzing the expression trends of key genes and miRNAs individually, we found that the results were highly consistent with the performance of their respective gene clusters. Interestingly, the expression level of *Pcna* was generally higher than that of other genes, while miR-21-5p even reached twice the expression level of other members (Figure 6c,d). As mentioned earlier, miR-21-5p was upregulated during tissue regeneration, which may suggest its role as a fundamental regulatory factor for wound repair and growth. The high expression level ensures its stable negative regulation of target genes, maintaining the normal progress of biological processes. Figure 6e illustrates the correlations among miRNAs, mRNAs, and the various time points.

## 3. Discussion

Salamanders, as a distinct group of vertebrates, exhibit remarkable tissue regeneration capabilities. Historically, significant strides have been made in studying the axolotl (*Ambystoma mexicanum*), exploring tissue regeneration at the omics level, as well as physiological processes [33,34,35]. Furthermore, non-model organisms also prove valuable for investigating tissue regeneration. Compared with salamanders, newts possess a superior regenerative ability [36], primarily reflected in the fact that newts can regenerate tissues throughout their lifespan, whereas salamanders retain this ability only during early life stages [37,38]. Therefore, genetic research on newts may uncover the genetic basis of more unique tissue regeneration phenomena compared with salamanders. In this study, we present a comprehensive analysis of mRNA and miRNA expression patterns during the early stages of limb regeneration in the Chinese fire belly newt. Our research identified 24 significantly differentially expressed mRNAs and 20 miRNAs. Enrichment analysis revealed numerous biological processes intricately linked with limb regeneration. Notably, the *Cemip* gene, associated with the hyaluronan catabolic process category, emerged as a highly differentially expressed gene. Employing stringent criteria for strong mRNA-miRNA interactions, our analysis successfully matched the identified candidate key genes (*Cemip*, *Rhou*, *Gpd2* and *Pcna*) with their respective regulated miRNAs. These findings reinforce our understanding and elucidation of the early expression patterns during limb regeneration in newts.

The traumatic limb injury caused by external forces differs significantly, from programmed cell apoptosis during early embryonic development, involving numerous unconventional biological processes that regulate the growth of new tissues [39]. During the initial stages of limb regeneration, the primary focus is on wound healing, bud formation, and growth. Our GO enrichment results highlight cellular extravasation as the most significant biological process, enriched by *Bst1*, *Ccl25*, and *Crk* genes. Cellular extravasation, the process of cells escaping from blood vessels or tissues, typically occurs in situations of inflammation, infection, or injury, where immune cells such as leukocytes [40] enter surrounding tissues through blood vessel walls to participate in immune responses or tissue repair. This crucial biological process aids the body in fighting infection or injury, promoting tissue regeneration and repair [35,41,42]. Previous studies on the mentioned genes have successfully validated their significant role in regulating cell adhesion at different levels, controlling the timing and rate of cell migration [23,43,44]. Consistent with expectations, we identified the regulation of cell-substrate adhesion in the yellow module, which includes cellular extravasation. These results suggest that in the early stages of limb regeneration, cell adhesion may play an important role in wound healing and bud formation. Simultaneously, cellular extravasation interacts with immune responses and inflammatory reactions to regulate the process. Interestingly, our research findings show that the clusters in the module negatively correlated with time-related traits are all immune-related biological processes (Figure 2). The relatively low immune capacity can maintain macrophages [40,45] at a relatively stable stage, preventing them from excessively attacking the injured limbs and ensuring the programmed differentiation and proliferation of cells.

*Cemip*, a gene present among significantly DE genes and associated with enriched biological processes related to hyaluronan catabolism, stands out as a highly intriguing and promising gene for further exploration. *Cemip*, plays a crucial role in encoding hyaluronic acid (HA), a major component of the extracellular matrix (ECM) [46]. Research has revealed that *Cemip* is regulated by the Wnt/β-catenin and TGF-β signaling pathways [26,47]. These pathways, extensively studied and confirmed, play crucial roles in embryonic development, cell cycle regulation, and cell proliferation [14,17,48,49]. Hyaluronic acid, a major component of the extracellular matrix, is widely recognized as a natural substance bound by *Cemip*. HA creates a moist environment that promotes cell growth and exhibits anti-inflammatory and antibacterial properties, thereby significantly contributing to the wound healing process [46,50]. Two down-regulated genes, *Rhou* and *Gpd2*, are noteworthy. *Rhou*, encoding a Ras family protein and belonging to the Rho GTPase family, regulates various signaling pathways critical for cell processes such as morphology, migration, adhesion, and matrix remodeling [30,51]. It also plays a regulatory role in the Wnt signaling pathway [52]. During the early stages of limb regeneration, *Rhou* primarily participates in the processes of cell proliferation and differentiation. The downregulation of this gene ensures that cells suspend migration and maintain a relatively primitive morphology while simultaneously reducing cell adhesion to promote proper alignment and organization of cells [53]. On the other hand, the downregulation of *Gpd2* can effectively diminish mitochondrial activity, resulting in a reduced generation of reactive oxygen species (ROS) [31]. Additionally, it modulates the oxidative metabolism pathway within the cell. This metabolic shift facilitates the provision of the essential energy and biosynthetic precursors required for rapidly dividing cells during the process of tissue regeneration. Consequently, the downregulation of *GPD2* supports the proliferation and survival of progenitor or blastemal cells [54], which are crucial for the successful regeneration of tissues. Interestingly, our WGCNA and GO results show that immune-related processes also exhibit a negative correlation with the early stage of regeneration, which supports the hypothesis that a mild inflammatory response and a lower level of immunity, both regulated by genes, interact to provide a relatively stable environment for tissue regeneration. Our analysis, particularly qPCR, demonstrated a periodic pattern of *Pcna* expression This gene is highly capable of mediating double-strand breaks (DSB) and directing their repair, a process essential for DNA repair, cell cycle regulation, and potentially closely linked to longevity and aging [55,56]. Salamander *Pcna* has been shown to evolve at an unprecedented rate among vertebrates, according to the previous study, which may be associated with their unique regenerative ability [57]. Given the Chinese fire belly newt’s remarkable regenerative capacity, periodic expression of *Pcna* may be crucial for the regeneration process.

In addition to the four genes we identified, exploring the miRNAs that target them is crucial for a comprehensive understanding of the molecular mechanisms involved. Notably, miR-21-5p emerges as a prominent candidate, exhibiting a significant increase in expression during limb regeneration in Mexican axolotls [2,58,59,60]. The expression level results mentioned earlier also indicated that the expression level of miR-21-5p was significantly higher than other candidates. Our findings further support this observation, strengthening the link between miR-21-5p and the regenerative process. This miRNA is frequently implicated in cancer research, serving as a detectable target for various cancer types [61,62,63]. Elevated expression levels of miR-21-5p are indicative of cancer, suggesting potential molecular parallels between cancer and tissue regeneration that merit exploration. Among the other key miRNAs (miR-19b-1, miR-19b-2, miR-127-5p, miR-150-5p, miR-194-5p, and miR-210-5p) we discovered, their primary function remains a class of cancer detection markers [64,65,66,67]. While current miRNA research primarily centers on their target genes, the unique molecular mechanisms of miRNAs pose challenges for direct functional studies. Given that one miRNA can target multiple genes, constructing an mRNA-miRNA regulatory network emerges as an effective research approach. As a result, the interaction landscape we have constructed makes the relationship between the two clearly visible. The existence of multiple targeted nodes also serves as a priority for further exploring molecular functions, providing insights into the future exploration of limb regeneration.

## 4. Materials and Methods

### 4.1. Sample Preparation and Transcriptome Sequencing

We performed right forelimb amputations on Chinese fire-bellied newts (*Cynops orientalis*) at the midstylopod level. Proximal healing tissues were collected at 0 hours, 1 day, 5 days, 10 days, and 20 days post-amputation. corresponding to the early stages of regeneration, which include wound healing, blastema formation, and differentiation process. Three biological replicates were conducted at each time point. RNA extraction was carried out using the Trizol protocol (Invitrogen, Carlsbad, CA, USA). Total RNA concentration and integrity were assessed using agarose gel electrophoresis, a NanoPhotometer spectrophotometer (IMPLEN, Westlake Village, CA, USA), and an Agilent Bioanalyzer 2100 system (Agilent Technologies, Santa Clara, CA, USA). cDNA libraries were constructed and subsequently sequenced on an Illumina HiSeq4000 platform (Illumina, San Diego, CA, USA), generating approximately 4 Gb of raw data for each transcriptome. miRNA libraries were constructed with a small RNA Sample Pre Kit and subsequently sequenced on an Illumina HiSeq2500 platform (Illumina, San Diego, CA, USA), generating approximately 10 Mb raw reads for each sample.

### 4.2. Construction of RNA-Seq Assembly and Counts Matrix of Expression

The sequencing data underwent quality control using FastQC v0.12.0 (https://www.bioinformatics.babraham.ac.uk/projects/fastqc/, accessed on 18 August 2024), and adaptor contaminants were filtered using Cutadapt v4.4 (https://cutadapt.readthedocs.io/en/stable/, accessed on 18 August 2024). Subsequently, transcript sequences were assembled using Trinity v2.15.1 [68]. To achieve higher-quality transcript sequences, TransPS v1.1.0 [69,70] was employed to perform re-scaffolding on the assembly data obtained from Trinity. We selected the axolotl (*Ambystoma mexicanum*) as a closely related species and utilized its protein sequence data for alignment and de-redundancy through the BLAST+ v2.2.26 [71]. Transcript abundance for each sample was estimated in a genome-free manner using Kallisto v0.50.0 [72], and an expression matrix of counts was constructed. To enhance the robustness of our analysis, we filtered out lowly expressed transcripts (count < 30) and explored relationships among all samples and biological replicates to eliminate potential confounders. For the functional annotation of the transcripts, we utilized Trinotate v4.0.2 (https://github.com/Trinotate/Trinotate/, accessed on 18 August 2024), which enabled the addition of functional annotations to facilitate subsequent analysis.

### 4.3. Identification of Known and Novel miRNAs

Quality control on miRNA reads was performed using the FastQC and Cutadapt, as previously described. The clean reads were aligned with the miRbase v22 database [73] using Bowtie v1.3.1 [74]. Subsequently, miRDeep2 v2.0.1.3 [75] was used to identify both known and novel miRNAs. miRNAs with read count greater than 5 and a true positive probability exceeding 60% were selected for further analysis. A miRNA was considered reliable if it expressed in at least two out of the three individuals at a given time point.

### 4.4. Differential Expression of mRNAs and miRNAs

Differential expression (DE) analyses were conducted on both mRNA and miRNA matrices across the time course using R package DESeq2 v1.40.2 [76] and edgeR v3.42.4 [77]. We assessed the expression patterns in blastemal samples at each time point relative to the 0-hour control. Genes were deemed significantly expressed if they exhibited a log2 fold-change (LFC) of ≥2 with a *p*-value < 0.05. For miRNA, due to their overall stable expression levels and relatively small data set compared to transcripts, significance was assigned to those with an LFC of ≥1 and a *p*-value < 0.05.

### 4.5. mRNA Co-Expression Network Analysis (WGCNA)

The R package WGCNA v1.72-1 [78] was utilized for mRNA co-expression network analysis, with ‘Time course’ designated as a phenotypic trait. Following a comparison of correlations among different samples, we proceeded with additional analyses using the recommended power β (soft threshold) of 28. Subsequently, gene modules were identified by evaluating the degree of gene expression changes and correlations over the time course. Each color signifies a module comprising a cluster of genes grouped together based on their expression patterns and associations.

### 4.6. Gene Ontology Enrichment Analysis

To conduct a gene ontology (GO) enrichment analysis, we employed the R package clusterProfiler v4.8.2 [79]. We utilized the org.Hs.eg.db v3.17.0 (https://bioconductor.org/packages/release/data/annotation/html/org.Hs.eg.db.html, accessed on 18 August 2024) package in R as the reference database. Our main emphasis was on gene ontology (GO) terms associated with “Biological Process (BP)”. The significance of overrepresentation was assessed based on the y FDR-adjusted *p*-value (<0.05).

### 4.7. Prediction of miRNA Target Genes and Identification of Potential miRNA–Gene Pairs

To elucidate the target genes of DE miRNAs, we utilized the R package multiMiR v1.22.0 [80], which integrates multiple microRNA-target databases. Bidirectional predictions were conducted using either DE genes or DE miRNAs as input to validate our findings. Subsequently, to identify negatively correlated miRNA–mRNA pairs, we performed a Pearson’s correlation test using DE miRNA and DE mRNA data. MiRNA–mRNA pairs were selected for downstream analyses if they exhibited a negative correlation (Pearson’s coefficient > 0.9). The interaction information was then imported into Cytoscape v3.8.0 [81] for visualization and further analysis.

### 4.8. Gene Expression Trend Analysis

To investigate the temporal changes in gene expression, we conducted a gene expression trend analysis using the R package Mfuzz v2.60.6 [82]. Given that each sample in our study comprised three biological replicates, we utilized the mRNA expression matrix with averaged values as the input, configuring the number of clusters to 8. For miRNAs, we used the same method, but the number of clusters was set to 4. Upon visualization, we were able to discern the temporal expression patterns of significantly DE genes, shedding light on their potential pivotal role in the limb regeneration process. To validate the expression trends across different time points, we conducted quantitative real-time PCR (qPCR) experiments. To improve accuracy, we introduced three biological replicates and two technical replicates for each time point.

## 5. Conclusions

In this study, we conducted an integrated analysis of mRNA and miRNA expression profiles at various time points during the early stages of limb regeneration in the Chinese fire-bellied newt. Cluster analysis and GO enrichment were employed, revealing that certain differentially expressed mRNAs are intricately associated with the regeneration process. Concurrently, we identified four candidate genes (*Cemip*, *Rhou*, *Gpd2* and *Pcna*) and seven miRNAs (miR-21-5p, miR-127-5p, miR-150-5p, miR-194-5p, miR-210-5p, miR-19b-1, and miR-19b-2) that regulate their expression. Furthermore, we explored their expression patterns at different time points during limb regeneration. These research findings illuminate the expression dynamics of mRNA and miRNA implicated in the genetic aspects of limb regeneration in newts, offering valuable insights for future investigations into the intricate biological processes underlying tissue regeneration.

## Figures and Tables

**Figure 1 ijms-25-09225-f001:**
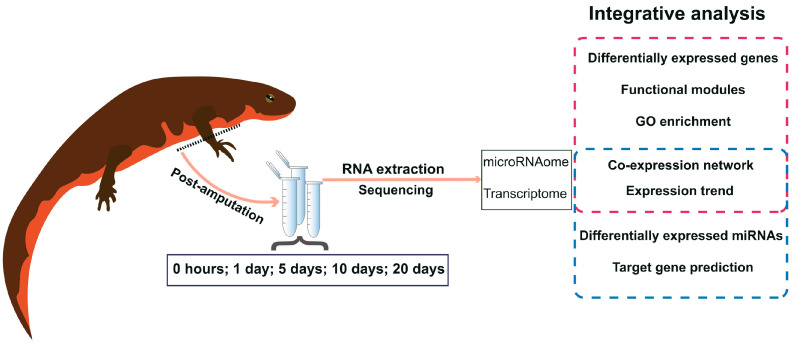
Schematic overview of the blastemal time course experiment and study strategy. Tissues collected from five distinct time points following the amputation experiment of the Chinese fire belly newt were utilized for RNA extraction. Subsequently, transcriptome and microRNAome sequencing was conducted. A comprehensive bioinformatics integration analysis was performed on these data, leading to the identification of key genes and targeting miRNAs that potentially exert significant regulatory functions during the early stages of newt limb regeneration.

**Figure 2 ijms-25-09225-f002:**
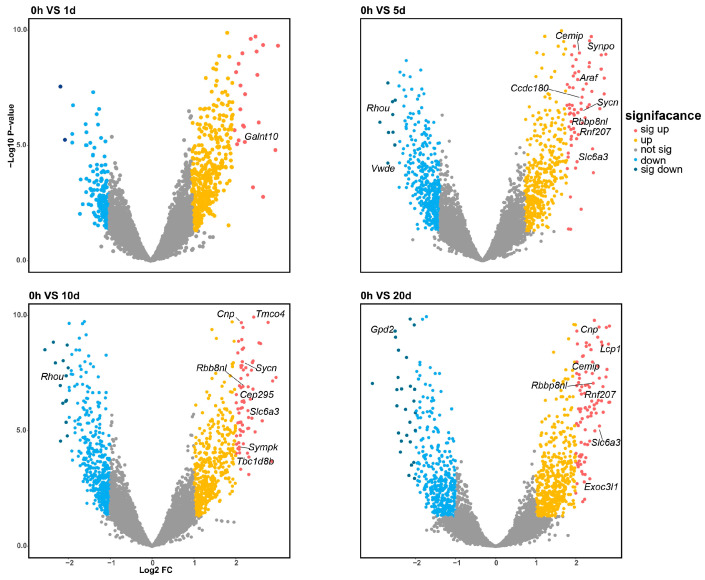
Four volcano plots illustrate the differential expression profiles of mRNA at 1, 5, 10, and 20 days post-amputation compared to the 0-hour control. The differentially expressed genes are color-coded according to their level of differential expression. The number of significantly DE genes increases with the time elapsed following amputation.

**Figure 3 ijms-25-09225-f003:**
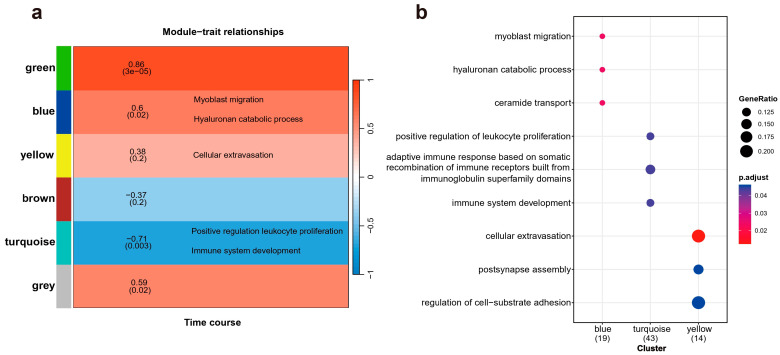
The relationship between gene module clustering and temporal characteristics, along with the biological processes enriched by these modules. (**a**) A total of six modules are annotated, with a gradient from red to blue indicating the varying strength of the correlation between each module and temporal traits. (**b**) A dot plot illustrating Gene Ontology (GO) enrichment displays all biological process entries surpassing the statistical significance threshold (*p* < 0.05).

**Figure 4 ijms-25-09225-f004:**
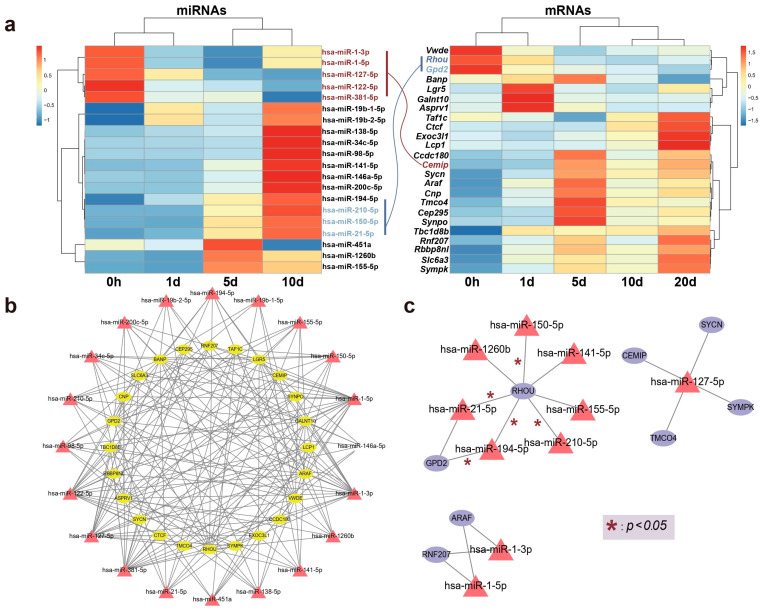
The relationship between candidate genes and miRNAs. (**a**) A heatmap illustrating the correlation between the relative expression levels of genes and miRNAs over time, with connections between key genes and miRNAs colored according to the direction of expression. (**b**) Establishment of a shared network expression relationship through the identification of interacting pairs. (**c**) A co-expression network of selected key candidate genes and their regulating miRNAs is illustrated, with edges marked with * denoting statistically significant interactions (* *p* < 0.05).

**Figure 5 ijms-25-09225-f005:**
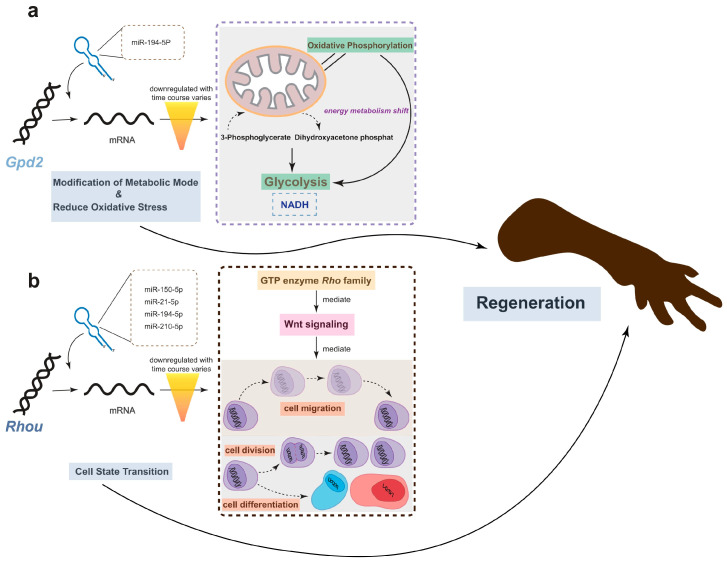
The potential effects of limb regeneration following the inhibition of expression of *Gpd2* and *Rhou*. (**a**) Inhibition of *Gpd2* affects mitochondrial aerobic metabolism, reducing NADH production efficiency through glycolysis, and mitigating the effects of oxidative stress. (**b**) As a member of the GTPase family, *Rhou* is involved in cytoskeleton dynamics and cell migration. The low expression of this gene ensures that cells cease migrating at the appropriate time and commence differentiation into the necessary cell types for regeneration. The combined low expression of both genes promotes regeneration.

**Figure 6 ijms-25-09225-f006:**
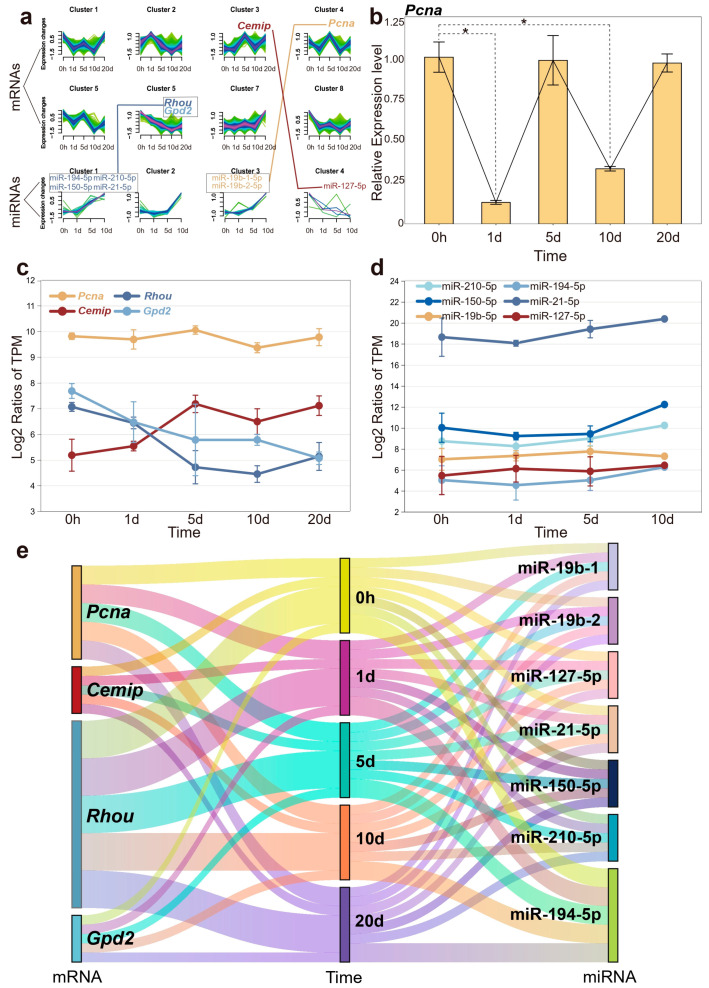
The expression dynamics of mRNA and miRNA over the time course. (**a**) Different clusters represent distinct expression trends of mRNA and miRNA, with connecting lines indicating their respective relationships. (**b**) Quantitative real-time PCR results for *Pcna*, with significant differences indicated (* *p* < 0.05). (**c**) The expression profiles of four candidate genes are depicted across different time points post-amputation. (**d**) The expression trends of pivotal miRNAs regulating candidate genes. (**e**) A river plot to illustrate the correlation among miRNAs, mRNAs, and time points. Each river represents a molecule, and the flow of these rivers visually demonstrates the interaction of mRNA and miRNA at different time points.

**Table 1 ijms-25-09225-t001:** Based on the module separation by WGCNA and the list of genes enriched by GO, distinct biological processes are enriched by the genes corresponding to each module, as outlined in the table.

GO ID	Module Color	Annotation	*p*-Value	Associated Genes
GO:0045123	Yellow	Cellular extravasation	2.13E-05	*BST1 CCL25 CRK*
GO:0070665	Turquoise	Positive regulation of leukocyte proliferation	3.89E-05	*BCL6 CSF1R JAK2 PTH SYK*
GO:0002460	Turquoise	Adaptive immune response based on somatic recombination of immune receptors built from immunoglobulin superfamily domains	7.51E-05	*BCH2 BCL6 C1QBP JAK2 SHLD2 UNG*
GO:0051451	Blue	Myoblast migration	7.65E-05	*NET1 SIX4*
GO:0002520	Turquoise	Immune system development	8.76E-05	*BCL6 ICOS SBDS SHLD2 UNG*
GO:0030214	Blue	Hyaluronan catabolic process	0.000117449	*CD44 CEMIP*
GO:0035627	Blue	Ceramide transport	0.000133028	*ABCA2 CPTP*
GO:0099068	Yellow	Postsynapse assembly	0.000403104	*CRK NLGN3*
GO:0010810	Yellow	Regulation of cell–substrate adhesion	0.000552828	*BST1 CCL25 CRK*

## Data Availability

Transcriptome and miRNA sequencing data were deposited in the Genome Sequence Archive (GSA) of the National Genomics Data Center (NGDC) at https://ngdc.cncb.ac.cn/gsa/ (accessed on 18 August 2024) under accession number PRJCA025436.

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
