# Peer review of "The mRNA and microRNA Landscape of the Blastema Niche in Regenerating Newt Limbs"

_ijms, 2024, doi:10.3390/ijms25179225_

Round 1
Reviewer 1 Report
Comments and Suggestions for Authors
In the manuscript titled "The mRNA and microRNA Landscape of the Blastema Niche in Regenerating Newt Limbs," the authors have explored the mRNA and microRNA profiles within the blastema niche of regenerating newt limbs, identifying critical genes and regulatory mechanisms involved in tissue regeneration. The study is compelling, and the authors have presented the data clearly and coherently. However, I have some corrections and suggestions to improve the quality of the article.
De Novo Transcriptome Assembly: The quality of the transcriptome assembly could be improved using a Transcriptome post-scaffolding (TransPS) and genomic reference-based re-assembly using a related species (e.g., the axolotl genome; https://www.axolotl-omics.org/assemblies), as demonstrated by Banerjee et al., 2019 (https://doi.org/10.1016/j.gene.2019.04.009) and Liu et al., 2014 (https://doi.org/10.1155/2014/961823).
Gene Names: The gene names should not be written in all caps, as this format is specific to human gene names. The authors should verify the nomenclature guidelines for this or any related species.
qPCR Validation: qPCR validation of some of the differentially expressed genes must be performed to confirm the expression patterns.
Figures: Correct the figure numbers. Increase the font size to improve the quality of the figures, as the current size makes them difficult to read.
Comments on the Quality of English Language
Overall, the manuscript was written coherently. However, minor corrections are necessary to improve quality.
Author Response
Dear reviewer,
Thank you very much for your comments and professional advice. Your insights have greatly contributed to the academic rigor of our article. Based on your suggestions, we have made the necessary revisions to the manuscript, and we hope that these changes meet your expectations.
Comment 1: In the manuscript titled "The mRNA and microRNA Landscape of the Blastema Niche in Regenerating Newt Limbs," the authors have explored the mRNA and microRNA profiles within the blastema niche of regenerating newt limbs, identifying critical genes and regulatory mechanisms involved in tissue regeneration. The study is compelling, and the authors have presented the data clearly and coherently. However, I have some corrections and suggestions to improve the quality of the article.
Response 1: We sincerely thank you for your positive assessment of our manuscript and for your valuable suggestions. We have carefully addressed each of the comments provided, as outlined below.
Comment 2: De Novo Transcriptome Assembly
The quality of the transcriptome assembly could be improved using a Transcriptome post-scaffolding (TransPS) and genomic reference-based re-assembly using a related species (e.g., the axolotl genome; https://www.axolotl-omics.org/assemblies), as demonstrated by Banerjee et al., 2019 (https://doi.org/10.1016/j.gene.2019.04.009) and Liu et al., 2014 (https://doi.org/10.1155/2014/961823).
Response 2: Thank you for this suggestion. We agree that enhancing the transcriptome assembly quality using the proposed methods could provide more robust data. We have incorporated a Transcriptome post-scaffolding (TransPS) approach and re-assemble the transcriptome using the axolotl genome as a reference. This method, as demonstrated by Banerjee et al., 2019 and Liu et al., 2014, will be instrumental in refining our assembly and improving the accuracy. The revised manuscript has included these updates.
Comment 3: Gene Names
The gene names should not be written in all caps, as this format is specific to human gene names. The authors should verify the nomenclature guidelines for this or any related species.
Response 3: We appreciate your attention to detail regarding gene nomenclature. We acknowledge that the current gene name formatting may be misleading. We have reviewed the Axolotl gene nomenclature guidelines (https://www.axobase.org/nomenclature) and ensured that all gene names are presented in the correct format. This correction has been applied throughout the manuscript to maintain consistency and accuracy.
Comment 4: qPCR Validation
qPCR validation of some of the differentially expressed genes must be performed to confirm the expression patterns.
Response 4: Regarding the qPCR validation of candidate genes, in previous experiments, possibly due to issues with primer design or experimental conditions, only the Pcna was successfully amplified and quantified, while the remaining candidate genes failed. The qPCR results for Pcna show a significant differential expression pattern, with a similar fluctuating expression trend to that observed in the transcriptome data. Due to the limited quantity of extracted RNA from various time points, most of which has already been exhausted, we were unable to perform additional qPCR validation. We apologize for any inconvenience caused by this issue.
Comment 5: Figures
Correct the figure numbers. Increase the font size to improve the quality of the figures, as the current size makes them difficult to read.
Response 5: Thank you for pointing out the issues with figure numbering and font size. The previous manuscript mistakenly labeled Figure 2 as Figure 1, and we have corrected this in the revised text. Additionally, we have increased the font size in the figures to enhance readability and ensure that the figures effectively communicate our results. These changes have been reflected in the revised manuscript.
Comments on the Quality of English Language: Overall, the manuscript was written coherently. However, minor corrections are necessary to improve quality.
Response: We appreciate your positive comments regarding its coherence. We have carefully reviewed the manuscript and made the necessary adjustments to enhance the quality of the language.

Reviewer 2 Report
Comments and Suggestions for Authors
In this manuscript, the authors investigated mRNA and miRNA expression during the regenerative stages in the fire-bellied newt, proposing novel genes and miRNA candidates involved in this process. In my opinion, the manuscript is well-supported by results, but it needs English revision and a reorganization of the results section. I propose to publish it after a minor revision, but with a significant reorganization of the manuscript.
Major Comments:
The paper is not easy to follow, especially the results section. In my opinion, the authors should report only the results obtained from the analyses in the “Results” section and avoid discussing them, as that information is repeated in the Discussion section. I suggest, if the journal allows it, to merge the two sections into a single section titled “Results and Discussion,” expanding on the missing part of the results.
The correlation between miRNAs and mRNAs and their differential modulation at the considered regeneration time points is not clearly represented. I suggest streamlining the text, adding a river plot to illustrate the correlation among miRNAs/mRNAs/time points, and perhaps including some graphs in supplementary material.
Abstract: Line 8: The work should be better introduced in the Abstract section by clearly explaining the focus of the research.
Figure 1: The figure, considering that it is in the introduction section, should only present the experimental plan steps and not the results of the research. I suggest removing the miRNAs and genes identified from the figure.
Section 2.1: Line 93: The sentence structure here is confusing. Furthermore, if all declared genes were differentially expressed at all time points (including 20 days post-amputation), the claim stating “…roles in the early stage of regeneration” is not justifiable. It may be useful to include a forelimb regeneration time-course to contextualize the treatment times within the scope of limb regeneration in this species.
Additionally, it is unclear to me why the authors discuss the PCNA gene in the results section, as the gene's result is not statistically significant. The relationship between the fast-evolving gene and regenerative capability is also unclear.
Line 115-116: The authors should explain the criteria chosen for group categorization.
Minor Comments: I found several formatting and style errors throughout the manuscript, such as:
Line 5: Incorrect affiliation style.
Line 15: A period is missing.
Line 72: A "dot" and a "comma" are both present simultaneously.
Line 110: Figure 2 is incorrectly referred to as Figure 1.
Line 17: “regulates” should be “regulate.”
Line 57: “Scholars” is not an appropriate term for identifying researchers.
Line 58-60: This sentence should be better formulated.
Line 67: "Newts" should be singular, and the scientific name is necessary.
In Fig. 1: “Nucleic acid extraction” could be replaced with “RNA extraction,” and among the reported treatment times, a symbol (e.g., a dash or semicolon) should be added.
Line 80: The sentence “The general idea and process of this study” should be improved for clarity.
Line 113: The sentence “escalate with the duration post-amputation” should be better articulated.
Lines 96-99: These lines should be moved to the Discussion section. In this manuscript, the authors investigated the mRNA and miRNA expression during the regenerative steps in the fire-bellied newt, proposing novel genes and miRNA candidates involved in such a process.
Comments on the Quality of English LanguageThe English language requires improvement. The authors often use non-scientific terminology throughout the manuscript.
Author Response
Dear reviewer,
Thank you for your valuable feedback and professional advice. Your insights have greatly contributed to enhancing the academic rigor of our article. Based on your suggestions, we have made the necessary revisions to the manuscript and hope that these modifications meet your expectations.
Comment 1: In this manuscript, the authors investigated mRNA and miRNA expression during the regenerative stages in the fire-bellied newt, proposing novel genes and miRNA candidates involved in this process. In my opinion, the manuscript is well-supported by results, but it needs English revision and a reorganization of the results section. I propose to publish it after a minor revision, but with a significant reorganization of the manuscript.
Response 1: Thank you for your thoughtful feedback on our manuscript. We appreciate your recognition of the strength of our results and your suggestions for improvement. We have carefully addressed the English language concerns and reorganized the results section as recommended. We believe these revisions will enhance the clarity and overall quality of the manuscript. Thank you for your constructive comments and for considering our work for publication.
Comment 2: Major Comments
The paper is not easy to follow, especially the results section. In my opinion, the authors should report only the results obtained from the analyses in the “Results” section and avoid discussing them, as that information is repeated in the Discussion section. I suggest, if the journal allows it, to merge the two sections into a single section titled “Results and Discussion”, expanding on the missing part of the results.
The correlation between miRNAs and mRNAs and their differential modulation at the considered regeneration time points is not clearly represented. I suggest streamlining the text, adding a river plot to illustrate the correlation among miRNAs/mRNAs/time points, and perhaps including some graphs in supplementary material.
Response 2: Thank you for your valuable feedback and constructive suggestions. We appreciate your insights, which have highlighted important areas for improvement in our manuscript.
We acknowledge that the “Results” section may have been challenging to follow due to the inclusion of interpretative content. Incorporating your suggestions and adhering to the journal’s formatting requirements, we have revised and deleted the additional discussions in the “Results” section of the manuscript, aiming to make the article more concise and accurate. Additionally, we have adjusted some text in the “Discussion” section to enrich the elaboration of the results.
We understand that the correlation between miRNAs and mRNAs across the different regeneration time points was not clearly represented. To address this, as per your suggestion, we have created a river plot (Fig. 6e) to visually illustrate the correlations among miRNAs, mRNAs, and the various time points. We believe this visual representation will provide a clearer understanding of the dynamic relationships involved.
Comment 3: Abstract: Line 8
The work should be better introduced in the Abstract section by clearly explaining the focus of the research.
Response 3: Thank you for your insightful feedback on our abstract. To provide more valuable information about this study in a clearer manner in the abstract, we have added a more intuitive research purpose to the “Abstract” section, making the logical flow of the entire abstract more explicit, with the hope of better showcasing the highlights of this article.
Comment 4: Figure 1
The figure, considering that it is in the introduction section, should only present the experimental plan steps and not the results of the research. I suggest removing the miRNAs and genes identified from the figure.
Response 4: Thank you for your thoughtful feedback. We have revised the Figure 1 accordingly to focus solely on the experimental plan steps and removed the miRNAs and genes identified. This adjustment will ensure that the introduction remains focused on outlining the research framework without prematurely presenting the results.
Comment 5: Section 2.1: Line 93
The sentence structure here is confusing. Furthermore, if all declared genes were differentially expressed at all time points (including 20 days post-amputation), the claim stating “…roles in the early stage of regeneration” is not justifiable. It may be useful to include a forelimb regeneration time-course to contextualize the treatment times within the scope of limb regeneration in this species.
Response 5: Thank you for your insightful feedback. We understand your concern regarding the sentence structure and the implications of the data in relation to the early stage of regeneration. To address this, we have included a limb regeneration time-course in the “Introduction” and “Materials and Methods” sections to better contextualize the treatment times. We hope this addition strengthens the interpretation of our findings and aligns with your suggestions.
Comment 6: Additionally, it is unclear to me why the authors discuss the PCNA gene in the results section, as the gene’s result is not statistically significant. The relationship between the fast-evolving gene and regenerative capability is also unclear.
Response 6: We appreciate your insightful comments. The inclusion of the Pcna gene in the results section was meant to provide context regarding cell proliferation in the regenerating blastema, as it plays a significant role in several cellular processes related to DNA repair and cell cycle regulation, and serves as a marker for cell proliferation. Salamander Pcna has been shown to evolve at an unprecedented rate among vertebrates, according to previous studies, which may be associated with their unique regenerative ability. In addition, our quantitative real-time PCR (qPCR) validations for Pcna demonstrated a periodic expression pattern that aligned with transcriptome data, but with significantly reduced expression at both 1 and 10 days post-amputation. The periodic expression of Pcna may be crucial for the regeneration process. We hope these revisions address your concerns, and we appreciate your constructive feedback, which has been invaluable in improving the clarity and quality of our manuscript.
Comment 7: Line 115-116:
The authors should explain the criteria chosen for group categorization.
Response 7: Thank you for your insightful comment. We acknowledge the importance of clearly explaining the criteria used for group categorization in the WGCNA analysis. In our revised manuscript, we have provided a detailed explanation of the criteria used to define the groups at the beginning of Section 2.2. We hope this clarification meets your expectations.
Minor Comments:
I found several formatting and style errors throughout the manuscript, such as:
Line 5: Incorrect affiliation style.
Response: Thank you for pointing out the issue with the affiliation style on line 5. We have corrected the format to align with the required guidelines.
Line 15: A period is missing.
Response: Thanks. We have added the missing period.
Line 72: A "dot" and a "comma" are both present simultaneously.
Response: Thank you for catching that error. We have deleted the comma.
Line 110: Figure 2 is incorrectly referred to as Figure 1.
Response: Thank you. We have revised it.
Line 17: “regulates” should be “regulate.”
Response: Thanks. We have corrected "regulates" to "regulate" in the revised Abstract.
Line 57: “Scholars” is not an appropriate term for identifying researchers.
Response:Thank you. We have changed "Scholars" to "Previous studies".
Line 58-60: This sentence should be better formulated.
Response: Thank you for your insightful feedback. We have revised the sentence to more clearly convey the importance of exploring limb regeneration with a focus on miRNAs, as suggested.
Line 67: "Newts" should be singular, and the scientific name is necessary.
Response: Thank you for your observation. We have corrected "Newts" to the singular form and included the appropriate scientific name in the revised manuscript.
In Fig. 1: “Nucleic acid extraction” could be replaced with “RNA extraction,” and among the reported treatment times, a symbol (e.g., a dash or semicolon) should be added.
Response: We have updated Fig. 1 by replacing "Nucleic acid extraction" with "RNA extraction" as suggested. Additionally, we have added a symbol to clarify the reported treatment times. We appreciate your valuable input in enhancing the clarity of our figure.
Line 80: The sentence “The general idea and process of this study” should be improved for clarity.
Response: Thank you for the feedback. To enhance its clarity, we have revised the sentence to “Schematic overview of the blastemal time course experiment and study strategy”.
Line 113: The sentence “escalate with the duration post-amputation” should be better articulated.
Response: Thank you. We have rephrased the sentence to “The number of significantly DE genes increases with the time elapsed following amputation”.
Lines 96-99: These lines should be moved to the Discussion section.
Response: Thank you for the suggestion. We agree that these lines would be more appropriately placed in the “Discussion” section as they seem to contribute more to the interpretation and analysis of the results rather than just presenting them. We have moved them accordingly to ensure the flow of the paper is consistent.
Comments on the Quality of English Language: The English language requires improvement. The authors often use non-scientific terminology throughout the manuscript.
Response: Thank you for your valuable suggestion. We acknowledge the need for refinement in the language used throughout the manuscript. We have carefully reviewed the text to ensure that scientific terminology is consistently applied and that the language aligns with the standards expected in academic writing. Your comments will help us enhance the clarity and precision of our manuscript.

Round 2
Reviewer 1 Report
Comments and Suggestions for Authors
The authors have significantly improved the quality of the manuscript.
Reviewer 2 Report
Comments and Suggestions for Authors
The authors have revised the manuscript to incorporate my suggested changes. In my opinion, the manuscript is now suitable for publication in its current form